# Wear Analysis of Additively Manufactured Slipper-Retainer in the Axial Piston Pump

**DOI:** 10.3390/ma15061995

**Published:** 2022-03-08

**Authors:** Agnieszka Klimek, Janusz Kluczyński, Jakub Łuszczek, Adam Bartnicki, Krzysztof Grzelak, Marcin Małek

**Affiliations:** 1Institute of Robots and Machine Design, Faculty of Mechanical Engineering, Military University of Technology, 2 Gen. S. Kaliskiego St., 00-908 Warsaw, Poland; agnieszka.klimek@wat.edu.pl (A.K.); jakub.luszczek@wat.edu.pl (J.Ł.); adam.bartnicki@wat.edu.pl (A.B.); krzysztof.grzelak@wat.edu.pl (K.G.); 2Institute of Civil Engineering, Faculty of Civil Engineering and Geodesy, Military University of Technology, 2 Gen. S. Kaliskiego St., 00-908 Warsaw, Poland; marcin.malek@wat.edu.pl

**Keywords:** additive manufacturing, wear analysis, mechanical properties, H13 tool steel

## Abstract

Additive manufacturing (AM) of spare parts is going to become more and more common. In the case of hydraulic solutions, there are also some applications of AM technology related to topological optimization, anti-cavitation improvements, etc. An examination of all available research results shows that authors are using specialized tools and machines to properly prepare AM spare parts. The main aim of this paper is to analyze the influence of quick repair of the damaged slipper-retainer from an axial piston pump by using an AM spare part. Hence, it was prepared with a 100-h test campaign of the AM spare part, which covers the time between damage and supply of the new pump. The material of the slipper-retainer has been identified and replaced by another material—available as a powder for AM, with similar properties as the original. The obtained spare part had been subjected to sandblasting only to simulate extremely rough conditions, directly after the AM process and an analysis of the influence of the high surface roughness of AM part on wear measurements. The whole test campaign has been divided into nine stages. After each stage, microscopic measurements of the pump parts’ surface roughness were made. To determine roughness with proper measurements, a microscopical investigation was conducted. The final results revealed that it is possible to replace parts in hydraulic pumps with the use of AM. The whole test campaign caused a significant increase in the surface roughness of the pump’s original parts, which was worked with the AM spare slipper-retainer: (1) from Ra = 0.54 µm to Ra = 3.84 µm in the case of two tested pistons; (2) from Ra = 0.33 µm to Ra = 1.98 µm in the case of the slipper-retainer. Despite significant increases in the surface roughness of the pump’s parts, the whole test campaign has been successfully finished without any damages to the other important parts of the whole hydraulic test rig.

## 1. Introduction

Additive manufacturing has recently caught the interest of many research teams. The mainstream of conducted research is still focused on technological and process issues. In the last two years, there is visible growth in the interest of using the AM in supply chains and spare parts analysis. This phenomenon was described by Frandsen et al. [1]. Earlier works were related to some exact applications or case studies analysis [2,3,4,5], however, for the most part, it has a form of preliminary analysis with that highlights the fundamental problems that arise during application of parts obtained using AM. Additionally, a lot of applied research is focused on some functional prototypes [6,7,8,9], in which the final parts are obtained using conventional manufacturing methods. Nowadays, trends in professional AM system development are mostly focused on the manufacturing of final products [10]. It is strictly related to a huge amount of research works connected with AM process analysis and data about the influence of different factors on the mechanical properties of produced parts [11,12,13,14,15,16,17,18].

From the other point of view, there is a significant growth in the interest of using AM technologies in supply chains. Verboeket et al. [19] indicated that AM technologies development allowed for growth in its direct usage for final products manufacturing. The authors highlighted that AM enhanced the potential of reaching lightweight, geometrically complex, and heavy-duty parts. Such an approach allows for a material usage reduction, lowering the fuel consumption in the AM parts in engines, reducing carbon footprints, and total production costs. Additionally, the same work [19] described the usage of AM in the context of spare parts/repair solutions in supplied chains. A usage of AM machines and local raw materials supplies in in situ and on-demand mechanisms resulted in the lower cost of transportation, raw materials, and finished goods. Such an approach improves responsiveness with improved product availability.

There is also a visible growth of interest in AM usage from an environmental point of view [20,21,22,23,24], and life cycle analysis [25,26]. An analysis made by Hapuwatte et al. in their work [27] indicated that using AM proves most sustainable in cases where geometrically complex components are necessary, however, they also highlighted that the quantity of production will be a deciding factor for considering AM as a production method.

Despite all the above-mentioned strengths of AM of final products and its usage in supply chains, the application of this technology in hydraulic solutions is still marginalized. There are only a few works related to the additively manufactured parts dedicated to hydraulic solutions [28,29,30,31]. The main reason for this phenomenon is wearing products generation, due to significant surface roughness directly after the process [32].

Continuous development of AM usage, especially in in situ shaping (which was a topic of our own research [33,34,35]) needs to be used in further work related to the specified application. Such an approach must be supported by life cycle assessment analysis of the hydraulic parts, described by Wang et al. [36], where the authors indicated the advantages of AM, especially from production maintaining and environmental points of view.

Hydraulic drives are very popular at heavy-duty machines like construction machinery, mining equipment, power plants, or the marine industry. The reasons why hydraulic drives are so popular are due to their enormous power (in comparison to mechanical or electric drives), high control possibilities, reliability, and flexibility.

The most important factor during the working equipment exploitation is reliability. Many research works are focused on improvements of commonly used parts, and their connections to the other part of the construction [37,38,39,40,41], however, a significant gap still remains in this field. It is very important to assure a proper schedule of the exact equipment exploitation, service, and diagnostic operations. That is why there is a need to provide the proper cooperation of resources, labor costs, and supply chains, which allows tasks to be accomplished within a specified deadline. The diagnostic systems installed on the machine, detailed observations, and analysis of the operating parameters allow for the flaws in identification before serious damage can occur.

Unfortunately, even the early detection of defects excludes the machine from its usage until a new part is ordered and replaced. Usually, the time of delivery of spare parts to the recipient takes about six weeks. Currently, due to the interruption or logistics chains issues, caused by the COVID-19 pandemic, the lead time can be longer. The situation when a damaged machine is out of service results in significant delays and penalties. The only solution to this problem seems to be the short-term rental of the replacement machine. Such an approach generates additional costs—for example—a 10 tons backhoe loader with the 70 kW engine power costs are about $1000.

According to the literature [42], approximately 20% of all damages in hydraulic systems are pump issues, which are the most important parts of the whole hydraulic system. Due to the importance of such parts, more effective diagnosing methods are in the interests of researchers [43,44,45]. One of the most popular types of hydraulic pumps is an axial piston pump with a variable capacity. Such pumps are characterized by high output power, high operating pressure, and compact design. The most common axial pump failures are damages of the slippers’ feet, pistons, the slipper-retainers, and the valve plates [46,47]. The damaged parts of an axial piston pump are shown in Figure 1.

There is also an additional factor that is very important in the case of hydraulic solutions, and at the same time, it is one of the biggest weaknesses of AM—the high surface roughness of the as-built parts [48,49,50,51,52,53,54]. In the case of H13 tool steel, Guenther et al. [52] revealed that in the case of such steels, a direct correlation between surface roughness and friction coefficient, i.e., the rougher the surface was, the higher the friction force, does not take place.

Additionally, in all the available research results related to the AM, spare parts for the hydraulic solutions are made with the use of specialized tools and machines to obtain a very high quality surface. In many cases, post-processing tools and machines (grinders, lathes, and milling machines) could be used to manufacture such parts conventionally without using AM. Those factors encourage authors of this work to analyze the slipper-retainer in an axial piston pump, especially from a wear resistance point of view, to check the possibility of maintaining the operation of such a part in the case of damage and lack of available spare parts. Additionally, to avoid complicated postprocessing activities, the AM spare slipper-retainer has been subjected only to sandblasting to remove sintered powder parts from the test part.

## 2. Materials and Methods

### 2.1. Material Type Detection

The test was investigated with the use of energy dispersive spectroscopy (EDS) and scanning electron microscope (SEM) Jeol JSM-6610 (Jeol Ltd., Tokyo, Japan). To allow more detailed material detection, Rockwell hardness distribution measurements were made using a W-1460 hardness tester (KABID-PRESS, Warsaw, Poland).

### 2.2. Manufacturing Process Description

The slipper-retainer of the test pump has been designed using reverse engineering based on simple caliper measurements covered by microscopical measurement verification. The test part (shown in Figure 2) was designed using CAD software (SolidWorks 2021, version number: 6.30.1030, Dassault Systèmes, Vélizy-Villacoublay, France) and manufactured using the SLM 125HL machine (SLM Solutions AG, Lubeck, Germany).

For the test part manufacturing, the default group of process parameters has been selected and specified in Table 1.

To avoid part deformations and warping phenomenon during the process (caused by a high-temperature gradient), the test part has been manufactured with a solid base (it was shown in Figure 3), which was cut out after the successful finish of the process using an Electro-Discharge Machine (EDM). Support structures were generated only below the angled surface of the downskin.

## 3. Experimental

### 3.1. Material Detection

The first part of the research was material identification, which is used to produce conventional parts (slipper-retainer from Hydraut PQ15 HLA2R S 40X axial piston pump). Based on the SEM-EDS analysis, the amount of each element in the analyzed material with its chemical composition is shown in Figure 4.

Additionally, to improve the accuracy of material type detection, Rockwell hardness distribution measurements were made. Measurement points have been distributed randomly on the flat surface of the original slipper-retainer. Obtained results are shown in Table 2.

Based on the chemical composition analysis and hardness measurements of the slipper-retainer, the material was specified as AISI A3145 steel. That type of material is mostly dedicated to manufacturing parts of engines and motor vehicles that need high impact resistance, good tensile strength, and ductility. Compared values of the analyzed factors are shown in Table 3. In the case of elements amount which were not able to measure, or detect instead of value —no data (nd) description was put. 

However, this steel in powder form is not offered by the suppliers on the market. To reach proper values of material properties, H13 tool steel has been selected as a substitute material, mostly due to its hardness at a level of 38 HRC without any additional heat treatment directly after selective laser melting (SLM), which is a Powder Bed Fusion (PBF) AM processing (in accordance with ISO/ASTM 52900).

The metallic powder (Carpenter Technology Corporation, Philadelphia, PA, USA) used for slipper-retainer replacement additive manufacturing was gas atomized H13 tool steel in the argon atmosphere. Obtained results using SEM, shown in Figure 5, revealed that powder particles had spherical shapes in a diameter of 15–45 µm.

The chemical composition of the used material is shown in Table 4. 

### 3.2. Hydraulic Tests Methodology

The main features of the prepared test were to preserve pump operating parameters and wear analysis. The tested pump was a Hydraut PQ15 HLA2RS40X (Hydraut, Via Lazzaretto, Italy). The main operating parameters of this pump are shown in Table 5.

In the field of hydraulic pumps, there is no clear procedure of durability tests, there are only “industry standards” provided by each company. Due to a lack of standards, an original test procedure was prepared. The total test time was set as 100 h—which is a typical three-week worktime regime [55]. This was based on the operating conditions presented in [55,56].

The test campaign consisted of five stages:0 stage (preliminary test), the pump was operating without any load for one hour, the operating pressure was generated only by losses in the hydraulic system.1st and 2nd stage, the pump was operating with a load of a 50, and 75 bars for eight h, respectively.3rd stage, the pump was operating with a load of about 100 bars for 16 h.4th stage, the pump was operating with a load of 125 bars for 32 h.5th stage, the pump was operating with a load of 150 bars for 35 h.

After each stage, the characteristic curves were measured. What is more, at each stage the pump was re-assembled, the slipper retainer and pistons were carefully checked, and the wear of the components was measured with an optical microscope.

### 3.3. Test Station

The test station was based upon a commercial (RDL, Miszewko, Poland) test rig, shown in Figure 6. The main parameters of hydraulic powerpack are 65 dm^3^ tank capacity, 900 rpm motor speed, and maximum power equal to 3 kW. The hydraulic scheme of the test rig is shown in Figure 6b. During the tests the oil temperature, flow and pressure were analysed. The oil temperature range was 40–46 °C.

### 3.4. Microscopic Investigation

The microscopic measurement and wear analysis had been made by using a Keyence VHX 7000 optical microscope (Keyence International, Mechelen, Belgium). The ability to tilt the measurement lens in the mentioned microscope allowed for wear analysis on the conical surface of the slipper-retainer part. The measurements related to the scratch depth were made using the cross profiles of the observed surface. The conical surface image was made using a 3D module (using the Keyence VHX 7000 microscope), which allowed a three-dimensional model creation of the tested surface. Subsequently, fifteen profile lines were made perpendicular to the direction of the scratches at 0.3 mm intervals. Such an approach allowed the creation of an average profile of the whole surface damage according to the base surface measurements. The roughness measurement was carried out in accordance with PN EN-ISO 4287 [57].

## 4. Results and Discussion

The main three parts of the tested pump, subjected to the highest wear (due to the cooperation with the AM slipper-retainer) were shown in Figure 7. Those surfaces are the spherical retainer guide (1 in Figure 7), spherical surface in the slipper-retainer (2 in Figure 7), and conical surfaces of the top parts of two selected pistons (3 and 4 in Figure 7). For each part of the pump, detailed microscopic measurements have been made.

Wear measurements of the tested pump parts related to R_a_ roughness are shown in Table 6. For each level—five measurements were made. To provide reliable data, statistical analyses were made in a form of standard deviation (Std. dev. in Table 6) and type A uncertainty (Uncert. A in Table 6) calculations.

Values compared in Table 5 are shown in the form of a columnar chart in Figure 8.

The visible high surface roughness of the spherical surface in the slipper-retainer is based on the obtained results, and is typical for parts manufactured during the SLM process and additionally subjected to sandblasting [35]. The significant surface roughness reduction (66%) is visible after exceeding the pump pressure value equal to 125 bar and kept by 32 h of exploitation. Further load increasing caused a slight drop of the R_a_ parameter of this part. In the case of the original parts (conical surface of the retainer guide and two pistons) there is visible an increase in the surface roughness, which is equal to 500% in the case of the retainer guide, and 555% in the case of the pump pistons. Such large growth was caused by the cooperation of the original parts with the AM slipper-retainer. After exceeding the mentioned total values, the surface roughness of three tested original pump parts started to decrease and reached the following values in comparison to the initial stage: retainer guide—172%, and pump pistons 267%.

Such a phenomenon is strictly related to different materials used for the production of the retainer guide (hardened steel) and pump pistons (tin bronze C90700). The significant wear took place in the case of pump pistons, which had been subjected to deeper microscope investigation based on geometrical analysis. The images from each stage are shown in Table 7.

Based on the register images (Table 6) and data shown in Figure 8, the different wear of two analyzed pistons after 1-h test without load could be observed. The most significant unevenness in the surface roughness of the conical pistons is visible in the first part of the test campaign.

In the case of the retainer guide surface roughness, there were no such significant changes after the whole test campaign. There were some irregular scratches, although not in the whole area of the conical surface of the retainer guide. After the last test stage (with the total load equal to 150 bars), a large defect occurred in the conical surface (Figure 9).

Taking into account the defect shape shown in Figure 9, one could conclude that such a defect has a fatigued character. Such a phenomenon is similar to Rolling Contact Fatigue (RCF) spalling effect, as a result of characteristic defect shape with a visibly scaly structure [58]. Due to the high roughness of the surface of the printed element, the lubricating film was broken and the cooperating surfaces were in direct contact. This effect was conducive to the cyclical increase in stresses and the material particles being pulled out. As a result, spalling craters were formed on the surface of the elements (Figure 9).

As mentioned, after each stage of the test the pump characteristics were measured. Significant wear of the piston surface (in a contact with the slipper-retainer) affected the movement of the pistons, which is crucial for the pump flow. Results are presented in Figure 10. The dotted lines represent the new pump characteristics, and the continuous lines represent the curves for the pump with the replaced slipper-retainer.

The measured curves for 100 and 125 bar are consistent with the new pump ones. The differences in runs are neglected and can be a result of system dynamic behaviors. A significant difference is observed with the curve for 150 bar—with the growing flow, the system pressure decreases. That phenomenon may be caused by serious defects in tested parts.

## 5. Conclusions

In this research, the SLM processing of the slipper-retainer without additional postprocessing was conducted. This allowed for successful restoration of the valve pump’s mobility. A 100-h test campaign with different load conditions has been successfully passed without any damage to the test rig. Such practical outcomes could be used for other research related to optimization design from both topological and hydraulic efficiency points of view.

Based on the obtained research results, the following conclusions could be drawn:The postprocessing of the AM parts, based only on the sandblasting process, allows for the successful exploitation of the pump in safe mode, however, at the same time, it is not sufficient to properly reduce the surface roughness. The further grinding (even manual) of the cooperating surface could significantly reduce the total wear.The surface roughness of the additive manufactured retainer guide has been decreased after exploitation in 125 bar of the pump load. Persisting the load on the level of 125 bars for 32 h allowed a reduction in the surface roughness of a value of 67%.After exceeding the load value above 125 bars (which is 45% of the nominal pressure), the surface roughness of the slipper-retainer spherical surface started to increase.Exceeding the total load value equal to 150 bars caused significant wear of the retainer guide, which has a spalling-like character.After 32 h of pump operation time with 125 bar pressure (total operating time 65 h), the pump operating parameters were consistent with theoretical ones. A rise in the system pressure (to 150 bars) resulted in significant wear. With the growing wear of the elements of the pump, the operation parameters deteriorated.Additive manufactured spare parts obtained with the use of PBF technologies dedicated to metallic powders allows the continuation of the exploitation of such devices as hydraulic pumps, until the arrival of a new pump delivery.Due to the significant wear of the elements, this solution is not recommended for fragile and precise systems. The debris in oil may have caused damages to the other parts. To avoid that problem, an additional filter in the system is recommended.

The outcomes from this research would be used for future work related to the generative design, and topological optimization of the parts dedicated for the hydraulic solutions. To maintain the practical approach, only simple postprocessing would be used (sandblasting, and manual grinding).

## Figures and Tables

**Figure 1 materials-15-01995-f001:**
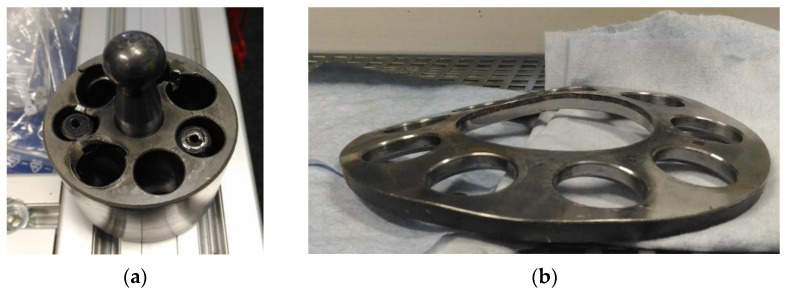
The damaged parts of an axial piston pump: (**a**) cylinder block with pistons, (**b**) slipper-retainer.

**Figure 2 materials-15-01995-f002:**
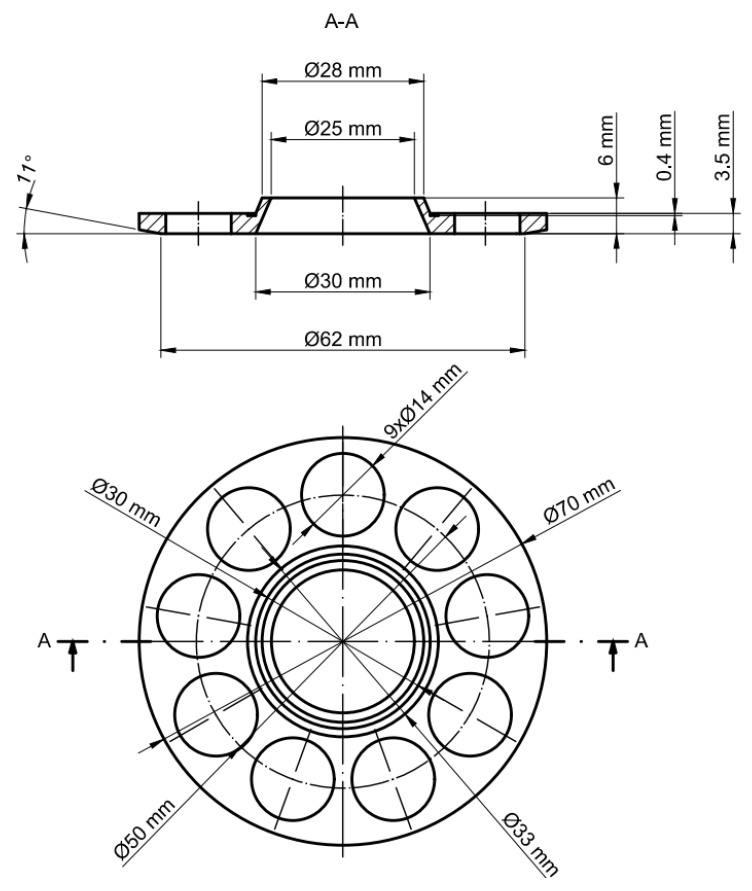
The slipper-retainer drawing with dimensioning.

**Figure 3 materials-15-01995-f003:**
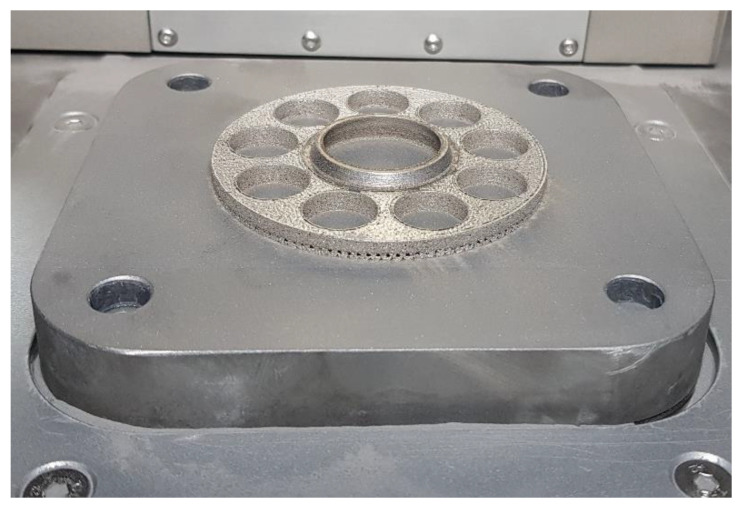
The slipper-retainer test part on the device’s building platform.

**Figure 4 materials-15-01995-f004:**
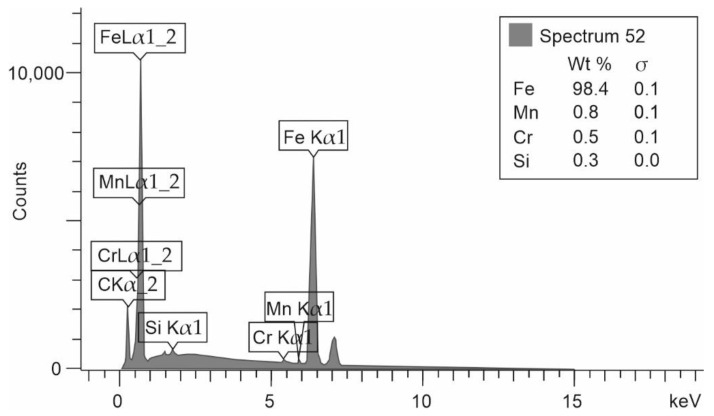
Chemical composition of analyzed slipper-retainer obtained in SEM-EDS analysis.

**Figure 5 materials-15-01995-f005:**
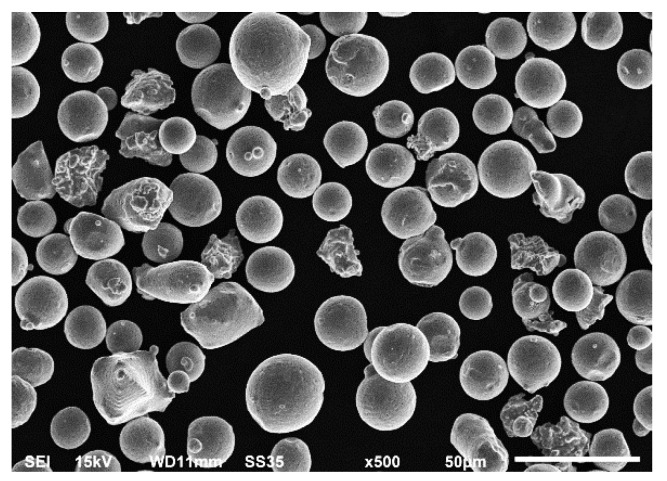
SEM image of H13L powder particles.

**Figure 6 materials-15-01995-f006:**
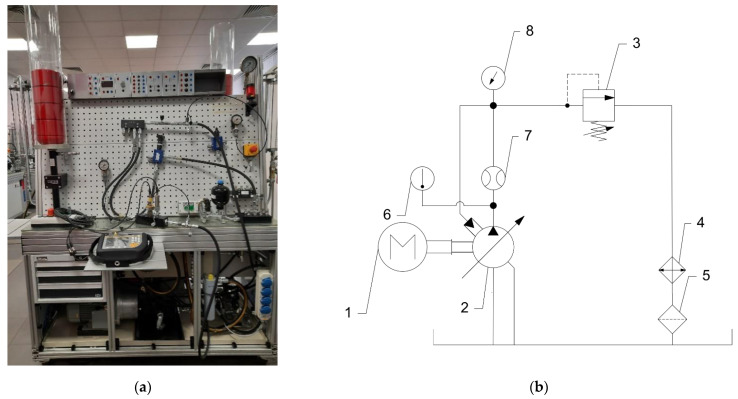
The test station: (**a**) station; (**b**) hydraulic scheme: 1—motor, 2—tested pump, 3—pressure relief valve (load valve), 4—cooler, 5—filter, 6—temperature sensor, 7—flowmeter, 8—pressure sensor.

**Figure 7 materials-15-01995-f007:**
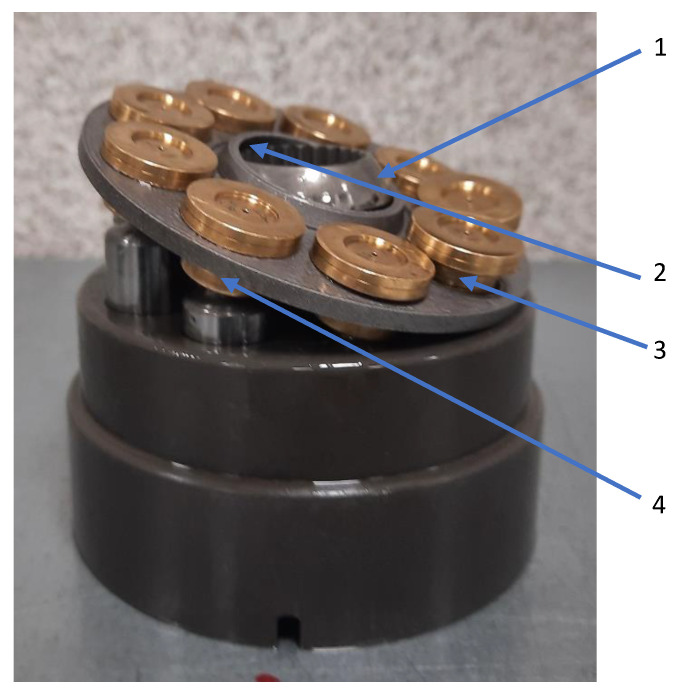
Four tested pump parts subjected to wear analysis: 1—spherical retainer guide; 2—spherical surface in the slipper-retainer; 3—the conical surface of the 1st piston; 4—the conical surface of the 2nd piston.

**Figure 8 materials-15-01995-f008:**
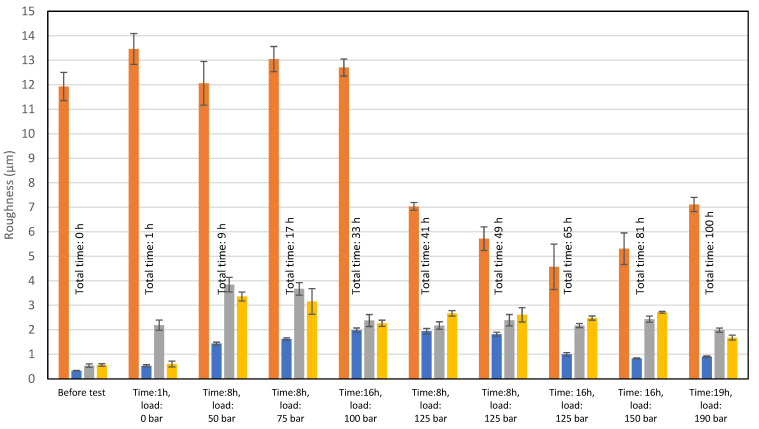
Roughness measurement results of all tested parts.

**Figure 9 materials-15-01995-f009:**
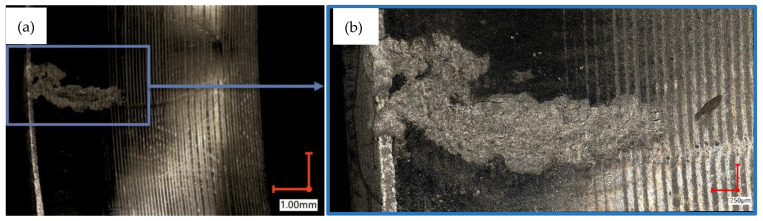
The hollow defect appeared after the last test stage (**a**)—×100 magnification; (**b**)—×300 magnification).

**Figure 10 materials-15-01995-f010:**
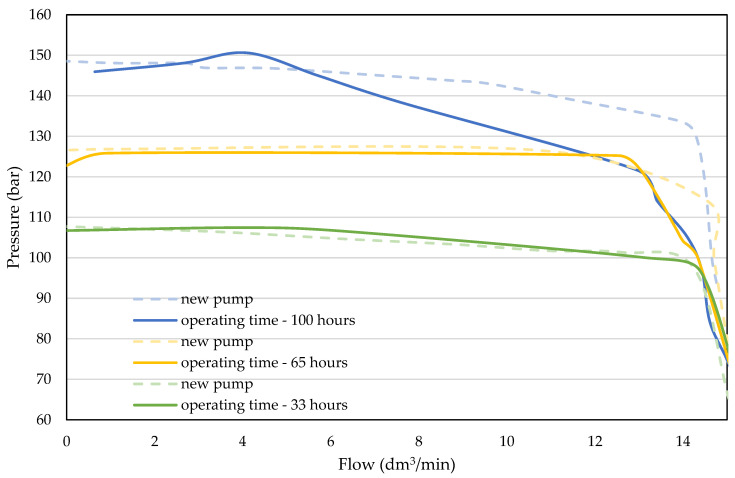
The pump characteristics.

**Table 1 materials-15-01995-t001:** Parameters used for samples manufacturing.

Layer Thickness l_t_ (mm)	Laser Power L_P_ (W)	Exposure Velocity e_v_ (mm/s)	Hatching Distance h_d_ (mm)	Energy Density ρ_E_ (J/mm^3^)
0.30	168	710	0.12	58.64

**Table 2 materials-15-01995-t002:** Hardness measurements of the slipper-retainer.

Measurement 1	Measurement 2	Measurement 3	Measurement 4	Measurement 5	Average	Standard Deviation
38 HRC	41 HRC	43 HRC	44 HRC	40 HRC	41 HRC	2 HRC

**Table 3 materials-15-01995-t003:** Comparison of the chemical composition and hardness analysis of the original material with the H13 tool steel.

**Chemical Composition (Nominal) Weight (%)**							**Hardness (Nominal) (HRC)**
C	Si	Mn	P	S	Cr	Ni	30
0.43–0.48	0.20–0.35	0.70–0.90	0.04	0.04	0.70–0.90	1.10–1.40	
**Chemical Composition (Measured) Weight (%)**							**Hardness (Measured) (HRC)**
C	Si	Mn	P	S	Cr	Ni	41
nd	0.3	0.8	nd	nd	0.5	nd	

**Table 4 materials-15-01995-t004:** H13 tool steel chemical composition.

C	N	S	P	V	Mo	Cr	Si	Mn	Cu	Ni
Weight [%]										
0.35	0.06	0.01	0.01	1.02	1.27	5.38	1.13	0.38	0.02	0.10

**Table 5 materials-15-01995-t005:** Hydraut PQ15 pump main parameters.

Max. Operating Pressure (bar)	Max. Displacement (cc/rev)	Max. Speed (rpm)
250	15	1800

**Table 6 materials-15-01995-t006:** Measured values of surface roughness in all tested pump parts.

Test No.	Before Test (µm)	Time: 1 h, Load: 0 bar (µm)	Time: 8 h, Load: 50 bar (µm)	Time: 8 h, Load: 75 bar (µm)	Time: 16 h, Load: 100 bar (µm)	Time: 8 h, Load: 125 bar (µm)	Time: 8 h, Load: 125 bar (µm)	Time: 16 h, Load: 125 bar (µm)	Time: 16 h, Load: 150 bar (µm)	Time: 19 h, Load: 190 bar (µm)
Spherical surface in the retainer guide										
1	0.32	0.46	1.22	1.62	1.89	1.52	1.74	1.03	0.78	0.95
2	0.31	0.45	1.45	1.79	2.09	2.22	1.87	0.89	0.89	0.84
3	0.35	0.51	1.56	1.53	2.29	2.00	1.66	0.88	0.87	1.00
4	0.37	0.7	1.41	1.61	1.81	2.06	1.71	1.26	0.82	0.85
5	0.32	0.55	1.53	1.55	1.82	1.89	2.10	0.92	0.78	0.87
Avr.	0.33	0.53	1.43	1.62	1.98	1.94	1.82	1.00	0.83	0.90
Uncert. A	0.01	0.05	0.06	0.05	0.09	0.12	0.08	0.07	0.02	0.03
Std. Dev.	0.02	0.09	0.12	0.09	0.18	0.23	0.16	0.14	0.05	0.06
Spherical surface in the slipper-retainer										
1	11.28	11.74	10.35	11.56	13.61	7.16	4.71	3.23	3.39	6.46
2	10.11	12.66	11.03	14.36	11.72	6.52	5.8	3.07	4.66	6.66
3	12.84	15.36	10.85	12.64	12.18	6.87	4.55	3.31	5.15	7.14
4	13.36	13.28	12.88	12.63	12.68	7.46	6.91	7.89	6.19	7.17
5	12.06	14.27	15.19	14.05	13.33	7.16	6.62	5.34	7.15	8.13
Avr.	11.93	13.46	12.06	13.05	12.70	7.03	5.72	4.57	5.31	7.11
Uncert. A	0.58	0.63	0.89	0.51	0.35	0.16	0.48	0.93	0.64	0.29
Std. Dev.	1.15	1.26	1.78	1.03	0.70	0.32	0.96	1.86	1.29	0.58
Conical surface of the 1st piston										
1	0.39	2.88	3.73	4.27	3.25	2.01	2.28	2.35	2.29	2.04
2	0.75	1.88	4.94	3.03	1.95	2.36	3.16	1.95	2.16	1.92
3	0.4	1.75	3.88	3.23	2.5	2.01	2.1	2.26	2.25	2.12
4	0.57	1.99	3.43	3.55	1.9	1.8	1.81	1.99	2.68	2.15
5	0.6	2.43	3.23	4.26	2.28	2.67	2.6	2.3	2.78	1.69
Avr.	0.54	2.19	3.84	3.67	2.38	2.17	2.39	2.17	2.43	1.98
Uncert. A	0.07	0.21	0.30	0.26	0.24	0.15	0.23	0.08	0.12	0.08
Std. Dev.	0.13	0.42	0.59	0.51	0.49	0.31	0.46	0.17	0.25	0.17
Conical surface of the 2nd piston										
1	0.56	0.7	2.76	5.09	2.53	2.9	3.4	2.67	2.73	1.57
2	0.65	0.36	3.36	3.15	1.91	2.78	2.08	2.61	2.62	1.41
3	0.69	0.31	3.45	3.07	2.02	2.36	2.21	2.4	2.85	1.99
4	0.45	0.88	3.9	2.27	2.32	2.44	2.12	2.51	2.67	1.81
5	0.47	0.77	3.32	2.19	2.53	2.87	3.23	2.19	2.7	1.61
Avr.	0.56	0.60	3.36	3.15	2.26	2.67	2.61	2.48	2.71	1.68
Uncert. A	0.05	0.11	0.18	0.52	0.13	0.11	0.29	0.08	0.04	0.10
Std. Dev.	0.09	0.23	0.36	1.05	0.26	0.23	0.58	0.17	0.08	0.20

**Table 7 materials-15-01995-t007:** The surface condition of two tested cylindrical surfaces of the pump pistons.

Piston 1		Piston 2	
Condition	Image	Condition	Image
Before test	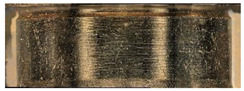		
Time:1 h, load: 0 bar	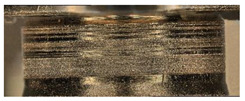	Time:1 h, load: 0 bar	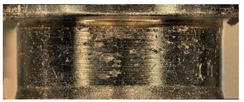
Time:8 h, load: 50 bar	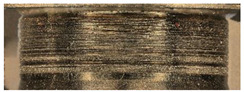	Time:8 h, load: 50 bar	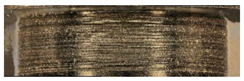
Time:8 h, load: 75 bar	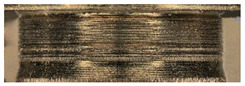	Time:8 h, load: 75 bar	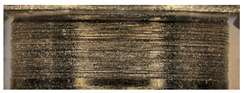
Time:16 h, load: 100 bar	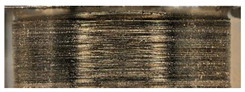	Time:16 h, load: 100 bar	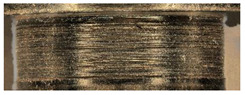
Time:8 h, load: 125 bar	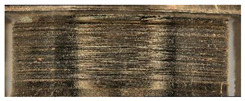	Time:8 h, load: 125 bar	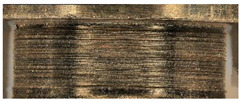
Time:8 h, load: 125 bar	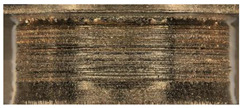	Time:8 h, load: 125 bar	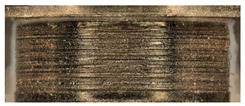
Time: 16 h, load: 125 bar	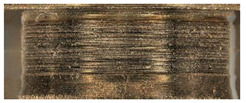	Time: 16 h, load: 125 bar	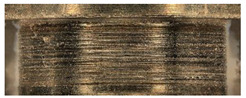
Time: 16 h, load: 150 bar	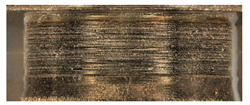	Time: 16 h, load: 150 bar	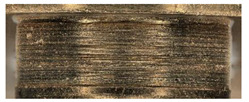
Time: 19 h, load: 150 bar	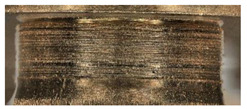	Time: 19 h, load: 150 bar	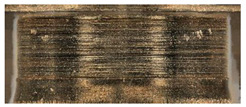

## Data Availability

Not applicable.

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
