# Peer review of "Wear Analysis of Additively Manufactured Slipper-Retainer in the Axial Piston Pump"

_materials, 2022, doi:10.3390/ma15061995_

Round 1

Reviewer 1 Report

Authors of this work have analyzed the slipper-retainer in axial piston pump, especially from a wear resistance point of view to check the possibility of maintaining the operation of that kind of part in the case of damage and lack of available spare parts. This manuscript requires major amendments. Please find comments as below:

  • Abstract should be restructured in light with best findings and contribution to the subject in the field.
  • Originality/novelty is expressed poorly which needs serious attention.
  • Why the originality is repeated in the material section?
  • Results are compromised in the methodology part.
  • Critical analysis should be added to the introduction section.
  • The procedure for SEM analysis and chemical composition should be included in the methodology section.
  • Quality of figures is very poor.
  • Results should be supported with valid info.
  • Error analysis is essential to be included.
  • Comprehensive proof read is necessary throughout the manuscript as there are a lot of typo/grammatical errors throughout the manuscript. 

Author Response

Reviewer 1

Dear Reviewer,

In the beginning, we would like to thank you for your revision and comment which were very helpful to improve our work. All corrections made in our manuscript which were made based on your comments have been yellow highlighted. Below you can find our answer to your comments.

Comment 1: Abstract should be restructured in light of best findings and contribute to the subject in the field.

Authors' response: We put the additional part at the beginning of our abstract: “Additive manufacturing (AM) of spare parts is going to be more and more common nowadays. In the case of the hydraulic solutions, there are also some applications of the AM technology related to topological optimization, any cavitation improvements, etc. In the case of all available results, authors were using specialized tools and machines to properly prepare AM spare parts.”

Comment 2: Originality/novelty is expressed poorly which needs serious attention.

Authors response:  The last part of the introduction has been extended to the following form: “There is also an additional factor that is very important in the case of hydraulic solutions, and at the same time, it is one of the biggest weaknesses of AM – the high surface roughness of the as-built parts. In all available research results related to the AM spare parts for the hydraulic solutions are made with the use of the specialized tools and machines to allow obtaining very high quality of the surface. In many cases, post-processing tools and machines (grinders, lathes, and milling machines) could be used to manufacture such parts conventionally without using AM. Those factors encourage authors of this work to analyze slipper-retainer in axial piston pump, especially from a wear resistance point of view to check the possibility of maintaining the operation of that kind of part in the case of damage and lack of available spare parts. Additionally, to avoid complicated postprocessing activities, the AM spare retainer-slipper has been subjected only to sandblasting to remove sintered powder parts from the test part.”

Comment 3: Why the originality is repeated in the material section?

Authors' response: Please accept our apologies. Before the submission, we have changed the template, and we have pasted the nonproper part. Now it is correct.

Comment 4: Results are compromised in the methodology part.

Authors' response: We have rephrased a part of this chapter to make it clearer. Now it has the following form: “The main features of the prepared test were to preserve pump operating parameters and wear analysis. The tested pump is a Hydraut PQ15 HLA2RS40X unit. The main operating parameters of this pump are shown in Table 4.”

Comment 5: Critical analysis should be added to the introduction section.

Authors' response: We were not able to form a critical analysis in the introduction part, because we were not able to find similar research of any hydraulic part in the available literature. The only thing we could do was to highlight the novelty of our work, which has been put at the end of our introduction – it has been yellow highlighted.

Comment 6: The procedure for SEM analysis and chemical composition should be included in the methodology section.

Authors' response: You are right, this part has been moved according to your advice.

Comment 7: Quality of figures is very poor.

Authors' response: We have saved our images in a 300p quality, the only one – EDS analysis has been damaged during data saving – we replaced it.

Comment 8: Results should be supported with valid info.

Authors' response: As it has been mentioned in our work there is no clear procedure for durability tests, there are only “industry standards” provided by each company. Due to a lack of standards, an original test procedure was prepared. All obtained results are already attached to our manuscript.

Comment 8: Error analysis is essential to be included.

Authors' response: Apart from standard deviation we attached a row in table 5 with a type A uncertainty. It was also included in a chart in figure 8.

Comment 9: Comprehensive proofread is necessary throughout the manuscript as there are a lot of typo/grammatical errors throughout the manuscript.

Authors' response: Our manuscript has been subjected to English editing, we put proper corrections in the text.

Reviewer 2 Report

This paper aims to replace the material of retainer – slipper in axial piston pump (AISI A3145 steel) with H13 tool steel manufactured by additive manufacturing technique. The current manuscript is interesting. It considered as an application of the reverse engineering approach. It can be accepted after dressing the following comments:

  • Moderate English changes are required in the revised manuscript

Abstract

  • The abstract is poorly written. The main aim of the work not clear.
  • What is the idea to replace the used material of the retainer-slipper with a new one having similar material properties? Is it economical target? Or is it designed for easy manufacturing? It is known that the H13 tool steel exhibits high hardness but economically it may be more expensive. All these issues should be clarified.
  • The meaning of this sentence is not clear “100-hour test campaign caused about 500% increase in the surface roughness of the pump’s original part, which was cooperated with the AM spare retainer slipper, without any damages to the test system.” Please revised it. In addition, the used parameter for measuring the surface roughness (Ra or Rz) should be used here.

Introduction

  • The introduction section needs to be improved. The problem statement and the aim of the work should be added. Moreover, new and related articles of the current journal can be cited.

Material and experimental details

  • In the material subsection, the written paragraph is repeated, it is the same paragraph in introduction part. Please resolve this issue.
  • The material of the slipper-retainer isn’t characterised (the chemical composition of the used materials isn’t illustrated in the text). The Figure 2 isn’t mentioned anywhere.
  • How are the hardness values in table 1 improved the accuracy of material type detection?
  • In this part, the method of measuring the surface roughness and the used surface roughness parameters should be stated here.
  • The authors should add the criteria selecting the process parameters in Table 3.

Experimental results and analysis

  • Regarding Table 5, it is recommended to use the errors or uncertainty not standard deviation.
  • The microstructure by SEM and Optical microscopic tools of the new retainer – slipper should be added to explain the improvement in the wear.
  • How the authors measured the scratch depth? Figure 9 needs be corrected, the errors should be added to each measurement. In addition, in Table 6, high resolution with higher magnification images should be used to see the differences.
  • This paper lacks more discussions.

Author Response

Reviewer 2

Dear Reviewer,

In the beginning, we would like to thank you for your revision and comment which were very helpful to improve our work. All corrections made in our manuscript which were made based on your comments have been green highlighted. Below you can find our answer to your comments.

Comment 1: Moderate English changes are required in the revised manuscript

Authors' response: Our manuscript has been subjected to English editing, we put proper corrections in the text.

Comment 2: The abstract is poorly written. The main aim of the work not clear.

Authors' response:: The abstract has been improved, it has now the following form:

Additive manufacturing (AM) of spare parts is going to be more and more common nowadays. In the case of the hydraulic solutions, there are also some applications of the AM technology related to topological optimization, anti-cavitation improvements, etc. In the case of all available results, authors were using specialized tools and machines to properly prepare AM spare parts. The main aim of this paper is to analyze the influence of quick repair of the damaged retainer – slipper from an axial piston pump by using AM spare part. Hence, it was prepared a 100-hour test campaign of the AM spare part which covers the time between damage and supply of the new pump. The material of the retainer-slipper has been identified and replaced by another material – available as a powder for AM, with similar properties as the original. The obtained spare part had been sub-jected to sandblasting only to simulate extremely rough conditions, directly after AM process and analyze the influence of the high surface roughness of AM part on wear measurements. The whole test campaign has been divided into nine stages. After each stage, microscopic measurements were made. To determine roughness with proper measurements microscopical investigation was made. The final results revealed that it is possible to replace parts in hydraulic pumps with the use of AM. The whole test campaign caused a significant increase in the surface roughness of the pump’s original parts which was worked with the AM spare retainer-slipper:

  • from Ra=0.54µm to Ra=3.84 µm in the case of two tested pistons,
  • from Ra=0.33µm to Ra=1.98 µm in the case of the retainer guide.

Despite significant increases in the surface roughness of the pump’s parts, the whole test campaign has been successfully finished without any damages to the other important parts of the whole hydraulic test rig.”

Comment 3: What is the idea to replace the used material of the retainer-slipper with a new one having similar material properties? Is it economical target? Or is it designed for easy manufacturing? It is known that the H13 tool steel exhibits high hardness but economically it may be more expensive. All these issues should be clarified.

Authors' response: We put in the abstract the following sentence to clarify our point:

“The main aim of this paper is to analyze the influence of quick repair of the damaged retainer – slipper from an axial piston pump by using AM spare part. Hence, it was prepared a 100-hour test campaign of the AM spare part which covers the time between damage and supply of the new pump.”

Comment 4: The meaning of this sentence is not clear “100-hour test campaign caused about 500% increase in the surface roughness of the pump’s original part, which was cooperated with the AM spare retainer slipper, without any damages to the test system.” Please revised it. In addition, the used parameter for measuring the surface roughness (Ra or Rz) should be used here.

Authors' response: The mentioned sentence has been rephrased and the percentage value has been replaced with Ra values.

Comment 5: The introduction section needs to be improved. The problem statement and the aim of the work should be added. Moreover, new and related articles of the current journal can be cited.

Authors' response: We have extended the final part of the introduction, where we tried to better expose the novelty of our work. We put additional citations:

  1. Tao Peng, Yanan Wang, Yi Zhu, Yang Yang, Yiran Yang, Renzhong Tang, Life cycle assessment of selec-tive-laser-melting-produced hydraulic valve body with integrated design and manufacturing optimization: A cra-dle-to-gate study,Additive Manufacturing,Volume 36,2020,101530,ISSN 2214-8604,
  2. Wang, P.; Lei, Y.; Qi, J.F.; Yu, S.J.; Setchi, R.; Wu, M.W.; Eckert, J.; Li, H.C.; Scudino, S. Wear behavior of a heat-treatable al-3.5cu-1.5 mg-1Si alloy manufactured by selective laser melting. Materials (Basel). 2021, 14, 1–11.
  3. Li, H.; Chen, Z.W.; Fiedler, H.; Ramezani, M. Wear behaviour of n ion implanted ti-6al-4v alloy processed by selective laser melting. Metals (Basel). 2021, 11.
  4. Sanguedolce, M.; Zekonyte, J.; Alfano, M. Wear of 17-4 ph stainless steel patterned surfaces fabricated using selective laser melting. Appl. Sci. 2021, 11.
  5. Yang, Y.; Zhu, Y.; Khonsari, M.M.; Yang, H. Wear anisotropy of selective laser melted 316L stainless steel. Wear 2019, 428–429, 376–386.
  6. Guenther, E.; Kahlert, M.; Vollmer, M.; Niendorf, T.; Greiner, C. Tribological performance of additively manufactured aisi h13 steel in different surface conditions. Materials (Basel). 2021, 14, 1–10.
  7. Huang, B.C.; Hung, F.Y. Al2 o3 particle erosion induced phase transformation: Structure, mechanical property, and impact toughness of an slm al-10si-mg alloy. Nanomaterials 2021, 11.
  8. Liu, X.; Wang, K.; Hu, P.; He, X.; Yan, B.; Zhao, X. Formability, microstructure and properties of inconel 718 superalloy fabricated by selective laser melting additive manufacture technology. Materials (Basel). 2021, 14, 1–18.
  9. Zeng, D.; Lu, L.; Gong, Y.; Zhang, Y.; Zhang, J. Influence of solid solution strengthening on spalling behavior of railway wheel steel. Wear 2017, 372–373, 158–168.

Comment 6: In the material subsection, the written paragraph is repeated, it is the same paragraph in introduction part. Please resolve this issue.

Authors' response: Please accept our apologies. Before the submission, we have changed the template, and we have pasted the nonproper part. Now it is correct.

Comment 7: The material of the slipper-retainer isn’t characterised (the chemical composition of the used materials isn’t illustrated in the text). The Figure 2 isn’t mentioned anywhere.

Authors' response: Based on the other Reviewer’s comment, this section have been moved and now figure 2 is figure 4, the proper reference has been put in the text. The chemical composition of the original material is shown in the small table in figure 4.

Comment 8: How are the hardness values in table 1 improved the accuracy of material type detection?

Authors' response: It is another factor (apart from chemical composition) that allows to properly choose the substitute material.

Comment 9: In this part, the method of measuring the surface roughness and the used surface roughness parameters should be stated here.

Authors' response: Thank you for this comment. It has been replaced according to your advice.

Comment 10: The authors should add the criteria selecting the process parameters in Table 3.

Authors' response: Dear Reviewer, please see the text above the mentioned table – we wrote that those are default process parameters (based on the SLM Solution’s official profile).

Comment 11: Regarding Table 5, it is recommended to use the errors or uncertainty not standard deviation.

Authors' response: Apart from standard deviation we attached a row in table 5 with a type A uncertainty. It was also included in a chart in figure 8.

Comment 12: The microstructure by SEM and Optical microscopic tools of the new retainer – slipper should be added to explain the improvement in the wear.

Authors' response: In the case of the AM retainer – slipper there were not any interesting changes (H13 is the most wear-resistant material among the rest parts used in our research). We made such images as supplementary file – it was included in our response as pdf file. We think it would be better if we will not attach unnecessary data in our manuscript. 

Comment 12: How the authors measured the scratch depth? Figure 9 needs be corrected, the errors should be added to each measurement. In addition, in Table 6, high resolution with higher magnification images should be used to see the differences.

Authors' response: We made an extended description of the methodology with the use of the microscope: “The measurements related to the scratch depth were made using the cross profiles of the observed surface. The conical surface image was made using a 3D module (using the Keyence VHX 7000 microscope), which allowed a three-dimensional model creation of the tested surface. Subsequently, fifteen profile lines were made perpendicular to the direction of the scratches at 0.3 mm intervals. Such an approach allowed to creation of an average profile of the whole surface damage according to the base surface measurements.” In connection with figure 9, we decided to remove this chart because of very low values which have a significant measurement error. If we would add some higher magnification images in table 6 there will be a problem with proper interpretation of the total wear of the piston's surface. A present form allows to visualize the wear of the whole surface, but for more specified analysis there are roughness measurement results shown in figure 8.

Comment 13: This paper lacks more discussions.

Authors' response: The discussion part has been extended and green highlighted.

Reviewer 3 Report

The research topic is urgent, but the research methodology is inferior.

The introduction is very uncertain. It should be rewritten with the focus on Your particular problem, but not general issues. Nowadays, additive manufacturing is a well-adopted manufacturing method.

Please, use professional English editing AFTER improving the article. The language style and numerous word mistakes make it impossible to access the manuscript completely.

Please, carefully describe and fulfill the research and follow my text recommendations. I ask You to justify the material selection for AM process. Please, ground the retainer ring before repeating the test and focus on the properties of the retainer ring You manufactured, but not on the slippers supplied by the manufacturer.  I think, after proper retainer ring preparation You will get a positive result.

The content of the article does not coincide with the title. There is no analysis of the wear-friction performance of the retainer ring.

There are mistakes in the reference list.

In this condition, I will recommend the Editorial office reject the manuscript

Author Response

Reviewer 3

Dear Reviewer,

In the beginning, we would like to thank you for taking your time to revise our manuscript (especially for the pdf file with specified comments).  Your comments were very helpful to improve our work. All corrections made in our manuscript which were made based on your comments have been blue highlighted. Below you can find our answer to your comments.

Comments from the pdf file:

Comment 1: Very contraversive. Boeing, Airbus, SpaceX use many hundreds of AM-parts in their products

Authors' response: Such a statement has been created based on the previous citations [2-5], not a commercial state-of-the-art.

Comment 2: Too long sentence. No chance to understand

Authors' response: The mentioned part has been rephrased.

Comment 3: I recommend You to start the introduction with this. And then disclose other problems and challenges of the method

Authors' response: It was very important for us to put some general information about the AM process and to describe some logistics processes with the use of AM technologies. Based on the “from general to specific” we suggest leaving it in a present form, especially since other reviewers' suggestions would have lost sense.

Comment 4: Why? Parts from Figure 1 have mechanical damage, not wear

Authors' response: We rephrased the sentence. We wanted to highlight that such parts are characterized by a higher possibility of damage

Comment 5: Show the figure with HRC distribution

Authors' response: We put the information that measurement points have been distributed randomly so there is no need to put the whole image.

Comment 6: This figure is not mentioned in the text. It should be in the "results" section. Is this XRD? The purpose and the methodology of this examination should be carefully explained. You should also add carbon and nickel to the results table

Authors' response: We put an extended caption. The EDS technology could not detect the amount of carbon. Indeed, in the AISI A3145 steel, there is a nickel, but it was not detected in the EDS analysis.

Comment 7: Please, give the references in literature where the steel like A3145 was substituted by H13 or other like steel. Why You didn't use structural steel with more favourable characteristics for this application?

Authors' response: It is not possible to select the same material for the AM process because for such an approach there would be a need of ordering "on-demand powder” which is many times more expensive in comparison to buying “more commercial” powders. The material selection is not based on the literature but on our independent decision.

Comment 7: If You initially decided to use this material, what was the need of manipulations with determining the properties of original material?

Authors' response: Because we wanted to keep the properties of the original part as much as it was possible

Comment 8: Reduce the image (fig.3)

Authors' response: The Figure has been reduced.

Comment 9: Improve the table formatting (Table 3)

Authors' response: The table has been formatted according to the journal instructions in the template file

Comment 10: Was the water cooling used during cutting? Did You remove the heat-affected layer after the cutting?

Authors' response: The whole substrate plate with the “printout” has been kept in the water during the whole process.

Comment 11: You may consult with operator and ask for the time "from new to first failure", and to select the test duration from this concern

Authors' response: We did it before the tests – it is a three-week worktime regime – we put proper information and citation in the text.

Comment 12: Why You didn't use weight loss measurements? Please, describe carefully the procedure of measurements.

Authors' response: It has not been taken into account because we analyzed only some specified surfaces. It is a very valuable comment, we will use such a method in our future research.

 Comment 13: Was it anywhere mentioned in Your earlier articles? Please, give the reference

Authors' response: No, it was not. It is a commercial test rig from the RDL company. It was highlighted in the text.

Comment 14: Was it dependent on oil pressure of oil flow?

Authors' response: As we mentioned in the test description, we have changed the pressure values.

Comment 15: I recommend You to use Laser scanning confocal microscopy. It allows to measure the volume wear of complex shapes, ant to determine the maximum and average wear depth in aromatic mode

Authors' response: We have tried to use a confocal microscope, but it was not possible to merge such a cylidrycal and conical surface. The Keyence VHX 7000 has a 3d measurement module that is more precise in the analysis of geometrically complex parts. 

Comment 16: I think You should add the figure explaining the methodology of measurements. The description above is unclear

Authors' response: We put information in the text that the roughness measurement was carried out in accordance with PN EN-205 ISO 4287. The description has been rephrased.

Comment 17: Why You didn't study the structure of AM component? Why didn't You do the microhardness profiles to control the AM-ed component quality? I also think that stress-relief annealing would be beneficial in this case.

Authors' response: Dear reviewer, it was not the topic of our article.

Comment 18: Please explain, why did You do this conclusion (The main three-part subjected to the highest wear)

Authors' response: The main reason was cooperation with the AM retainer guide. We put proper information in the text.

Comment 19: According to the figure, it is unclear what is studied. Please, add the cross-section of the pump, and arrow all the studied surfaces. Why did You study only 2 pistons?

Authors' response: Based on the assembled parts image it is easy to analyze which of them were in contact. The additional cross-section drawing is already shown in figure 2. We analyzed only 2 pistons to reduce unnecessary data (all pistons were subjected to wear equally – results from two tested pistons were a “necessary minimum”.

Comment 20: What is the roughness of new component?

Authors' response: We do not know what do you mean in the statement of the “new component” roughness. You could find all measured roughness values in figure 8.

Comment 21: Is this enough to conclude, that a part You have manufactured is completely unserviceable? What is the roughness of new pump component at this stage of test?

Authors' response: Please be informed that our main aim was to assure a three-week pump operation. We did not compare it with the original parts to prove that AM is better. It is not. We wanted to prove, that such an approach is sufficient to assure the operation of the damaged part before the arrival of the new pump.

Comment 22: Why didn't You grind the AM-ed component before incorporating it with a piston pump?

Authors' response: We put a proper explanation at the end of the introduction (below figure 1).

Comment 23: Do pistons actually contact with the retainer ring?

Authors' response: Yes. The cylindrical surface of their top parts.

Comment 24: Why do You analyse the pistons? As it comes from the article title, You should analyse the wear of retainer ring!

Authors' response: We indicated that the significant wear took place in the case of pump pistons, which had been subjected to deeper microscope investigation based on geometrical analysis. Please also look at figure 8 – the wear of the slipper-retainer was the smallest one in comparison to the other parts which were in cooperation with this spare retainer guide.

Comment 25: Is it piston surface? Or it is a surface of a slipper made of bronze?

Authors' response: We have treated it as a piston surface.

Comment 25: Load? Or oil pressure?

Authors' response: This part has been removed based on the other reviewer's suggestion.

Comment 26: I think it is a mistake here. These two figures ( mark them as a and b) look completely different. Right image is not an enlargement of the left.

Authors' response: Thank you for this comment. Indeed there was a wrong image. Now it is correct.

Comment 27: Pump pressure-flow output mainly depend on the piston-cylinder interface, and the clearance between piston block and port-plate. How this is dependent on the slipper-retainer ring interface?

Authors' response: We put in the manuscript the following sentence: “As it was mentioned, after each stage of the test the pump characteristics were measured. Significant wear of the piston surface (in a contact with the retainer-slipper) affected the movement of the pistons which is crucial for the pump flow.”

Comment 28: Use superscript (in figure 11)

Authors' response: The superscript was used

Comment 29: As a result of this operation, all slippers get damaged and require to be replaced. So we need another 6 weeks to wait for spare parts?

Authors' response: Our statement cause the misunderstood. We rephrased it: “Additive manufactured spare parts obtained with the use of PBF technologies dedicated to metallic powders allow to continue exploitation of such devices as hydraulic pumps until the arrival of the new pump delivery.

 Comment 30: What is the way out? As for me, the result is completely predictable. For this no test is required.

Authors' response: It could be predictable for the people who are specialists in this area, but our main aim was to analyze the whole exploitation process and measure the influence of such a 100-hour test campaign.

General comments:

Comment 1: The research topic is urgent, but the research methodology is inferior.

Authors' response: We respect your comment, but we could not agree with your statement. As we mentioned in our manuscript: In the field of hydraulic pumps testing, there is no clear procedure of durability tests, there are only “industrial standards” provided by each company. Due to a lack of standards, an original test procedure was prepared. The total test time was set as 100 hours. For our tests, we used all necessary laboratory devices, but at the same time, we tried to keep the practical character of our work. We hope you will take into account our point of view.

Comment 2: The introduction is very uncertain. It should be rewritten with the focus on Your particular problem, but not general issues. Nowadays, additive manufacturing is a well-adopted manufacturing method.

Authors' response: As we have mentioned in the first words of our manuscript – we are aware of the well-adoption of the AM, that is why we are writing about such thing as a kind of “mainstream” in the present research related to the AM technologies. Based on other reviewers' comments we have rebuilt the introduction part. New parts have been highlighted. Our topic is not very popular so it is very difficult to find some similar works to put some specified knowledge, but we tried to better highlight the novelty and originality of our work at the end of the introduction part. Also, the abstract has been rewritten.

Comment 3: Please, use professional English editing AFTER improving the article. The language style and numerous word mistakes make it impossible to access the manuscript completely.

Authors' response: Our manuscript has been subjected to English editing, we put proper corrections in the text.

Comment 4: Please, carefully describe and fulfill the research and follow my text recommendations. I ask You to justify the material selection for AM process. Please, ground the retainer ring before repeating the test and focus on the properties of the retainer ring You manufactured, but not on the slippers supplied by the manufacturer.  I think, after proper retainer ring preparation You will get a positive result.

Authors' response: Your suggestion is related to the different research methodology which was not our point. We tried to explain it in the text: “Additive manufacturing (AM) of spare parts is going to be more and more common nowadays. In the case of the hydraulic solutions, there are also some applications of the AM technology related to topological optimization, anti-cavitation improvements, etc. In the case of all available results, authors were using specialized tools and machines to properly prepare AM spare parts. The main aim of this paper is to analyze the influence of quick repair of the damaged retainer – slipper from an axial piston pump by using AM spare part. Hence, it was prepared a 100-hour test campaign of the AM spare part which covers the time between damage and supply of the new pump. The material of the retainer-slipper has been identified and replaced by another material – available as a powder for AM, with similar properties as the original.”

Regarding the material selection, we also proper explanations in our manuscript: “However, this steel in powder form is not offered by the suppliers on the market. To reach proper values of material properties H13 tool steel has been selected as substitute material, mostly because of its hardness at a level of 38 HRC without any additional heat treatment directly after selective laser melting (SLM), which is a Powder Bed Fusion (PBF) AM processing (in accordance with ISO/ASTM 52900).” It is not possible to select the same material for the AM process because for such approach there would be need of ordering "on-demand powder” which is many times more expensive in comparison to buying “more commercial” powders.

Please be informed that all wear analyses were made of the pump with the AM spare part (retainer guide). Our test campaign was successful, so we do not treat our result as negative. Of course, we had some critical observations, but those conclusions would be certainly used in future research.

Comment 5: The content of the article does not coincide with the title. There is no analysis of the wear-friction performance of the retainer ring.

Authors' response: There are not any standards related to friction performance analysis of such parts as retainer guides, pistons, etc. We are open to your suggestions for the new title of our work.

Comment 6: There are mistakes in the reference list.

Authors' response: Thank you for that comment. We made proper corrections and improvements.

Comment 7: In this condition, I will recommend the Editorial office reject the manuscript

Authors' response: We hope our improvements based on your and other reviewers' comment would meet your expectations of the new form of our manuscript. 

Round 2

Reviewer 1 Report

Authors have addressed the comments properly and can be accepted in the current form. 

Author Response

Dear Reviewer, 

Thank you very much for all your suggestions and your positive decision. 

Sincerely, 

Authors

Reviewer 2 Report

The authors have addressed all comments. only minor revision is required to be fixed: please add the unit of the dimensions in Figure 1. The dimensions are in mm. 

Author Response

Dear Reviewer, 

Thank you very much for all your suggestions and your positive decision. Figure 1 has been improved. 

Sincerely, 

Authors

Reviewer 3 Report

Thank for Your work. Now the manuscript is appropriately designed. Thank You for the corrections Yoi did. Now I will recommend to publish it.

Author Response

(The authors gave the same response as above.)
